# Ordering, flexibility and frustration in arrays of porphyrin nanorings

Alex Summerfield[1], Matteo Baldoni[2,3], Dmitry V. Kondratuk[4], Harry L. Anderson [4], Stephen Whitelam[5], Juan P. Garrahan[1], Elena Besley[2] & Peter H. Beton [1]

The regular packing of atoms, molecules and nanoparticles provides the basis for the understanding of structural order within condensed phases of matter. Typically the constituent particles are considered to be rigid with a fixed shape. Here we show, through a combined experimental and numerical study of the adsorption of cyclic porphyrin polymers, nanorings, on a graphite surface, that flexible molecules can exhibit a rich and complex packing behaviour. Depending on the number of porphyrin sub-units within the nanoring we observe either a highly ordered hexagonal phase or frustrated packing driven by directional interactions which for some arrangements is combined with the internal deformation of the cyclic polymer. Frustration and deformation occur in arrays of polymers with ten sub-units since close packing and co-alignment of neighbouring groups cannot be simultaneously realised for nanorings with this internal symmetry.

[1] School of Physics and Astronomy, University of Nottingham, Nottingham NG7 2RD, UK. [2] School of Chemistry, University of Nottingham, Nottingham NG7 2RD, UK. [3] CNR-ISMN Bologna, Via Piero Gobetti 101, 40129 Bologna, Italy. [4] Department of Chemistry, University of Oxford, Oxford OX1 3TA, UK. [5] Molecular Foundry, Lawrence Berkeley National Laboratory, 1 Cyclotron Road, Berkeley, CA 94720, USA. Correspondence and requests for materials should be addressed to P.H.B. (email: peter.beton@nottingham.ac.uk)

The packing of spheres, disks and other simple shapes into arrays in two and three dimensions provides the foundation for the theory of crystallography, and is relevant to many problems across the physical sciences with characteristic length scales ranging from the atomic up to the macroscale[1–3]. For cases with high symmetry such as hard disks, the minimum energy configuration in two dimensions corresponds to hexagonal close-packing with six-fold co-ordination, which maximises van der Waals interactions. The packing of objects with reduced internal symmetry leads to interactions, which compete with a simple drive to hexagonal close-packing, and over recent years molecular self-assembly has provided a route to explore more exotic arrangements such as quasicrystals, random tiling and fractals[4–13]. While it is valid in many cases to assume that the shape of the object undergoing packing is fixed, additional complexity is introduced for deformable objects, which can modify their shape locally to maximise interactions with nearest neighbours. In particular, deviations from simple packing rules derived for rigid objects should occur when the elastic energy associated with shape deformation becomes comparable with other relevant energy scales, such as nearest-neighbour interactions[14,15].

Here, we describe the close-packed assemblies of cyclic porphyrin polymers, referred to as nanorings, which display a frustrated packing in which the internal symmetry competes with the drive to hexagonal order, and individual nanorings responsively modify their shape to maximise local interactions with nearest neighbours. We use a combination of ab initio, molecular dynamics (MDs) and coarse graining computational approaches to understand this packing behaviour and to show that similar patterns emerge in computer simulations of deformable nanorings. Both the theoretical and experimental observations show that significant differences occur for nanorings with and without six-fold symmetry in agreement with the expected phase behaviour of disks with sticky patches[16] as recently calculated within a statistical mechanics framework.

## Results

**Experimental studies of nanoring adsorption.** The nanorings, which have recently been synthesised using a directed self-assembly process[17–20], consist of an integer number, $N$, of Zn porphyrin units linked together by butadiyne spacers in a cyclic arrangement (see Fig. 1a; $c$-PN denotes a cyclic nanoring with N porphyrin groups). These materials, and their linear analogues, attract great interest since they are highly conjugated polymers in which the optical absorption and emission properties are strongly influenced by the number of porphyrin groups[21–24]. The cyclic variants also serve as synthetic, macromolecular analogues of light-harvesting complexes in photosynthetic membranes[21,25,26].

Nanorings may be immobilised on Au(111) surfaces either from solution, or using electrospray deposition, allowing structural characterisation by scanning tunnelling microscopy (STM)[18,20,27], but their limited diffusivity on Au(111) has constrained the formation of regular extended structures. However, as we show below, deposition of nanorings on graphite, a more weakly interacting substrate, which has been explored widely in the context of molecular self-assembly[28–30], allows the formation of large domains of close-packed and quasi-close-packed arrangements, which are stabilised by inter-nanoring interactions. This leads to complex modes of in-plane organisation, which are determined by the number of porphyrin groups and driven by a competition between nearest-neighbour interactions, variations of the internal symmetry, and the elastic energy associated with elastic deformation.

The nanorings incorporate alkane groups ($OC_8H_{17}$ sidechains) attached to the porphyrin macrocycles (see Fig. 1) to promote solubility of the nanorings in a range of solvents. We have deposited layers with monolayer thickness by immersion of a highly oriented pyrolytic graphite (HOPG) substrate into a 10 μg/ml toluene: methanol (39:1) solution of either $c$-P10 (Fig. 1b) or $c$-P12 (Fig. 1c), after which the surface was blown dry. Images of the surface are then acquired using either STM or atomic force microscopy (AFM). A drop of nonanoic acid is added to the surface during STM imaging so that images of the nanorings are acquired at the interface between a liquid and the HOPG substrate; this procedure resulted in more stable imaging conditions. AFM images are acquired for dried films under ambient conditions. Full details of the sample preparation and STM/AFM imaging procedures are given in Methods.

Figure 2 shows STM (a, b, d, e) and AFM (c, f) images of the cyclic polymers $c$-P12 (top row) and $c$-P10 (bottom row). Large area STM images of $c$-P12 (Fig. 2a) show extended hexagonally ordered domains with lateral dimensions of up to ~1 μm with a periodicity of $6.2 \pm 0.1$ nm between nanoring centres when imaged in liquid (nonanoic acid). In addition we observe a single rotational domain for the $c$-P12 array and, consistent with this observation, find that the lattice vectors of the nanoring array and the graphite are aligned (this is determined by varying the STM scanning parameters so that the underlying graphite lattice may be resolved as shown in Supplementary Information (SI; Supplementary Fig. 1).

A hexagonal array would be expected to arise for a close-packed arrangement of highly symmetric near-circular objects, since this configuration maximises the isotropic van der Waals interactions between neighbouring rings. However, the intramolecular structure of the nanorings imparts a reduced $N$-fold internal symmetry, which is evident in images of $c$-P12 with higher resolution (Fig. 2b); in these images, the 12 bright features

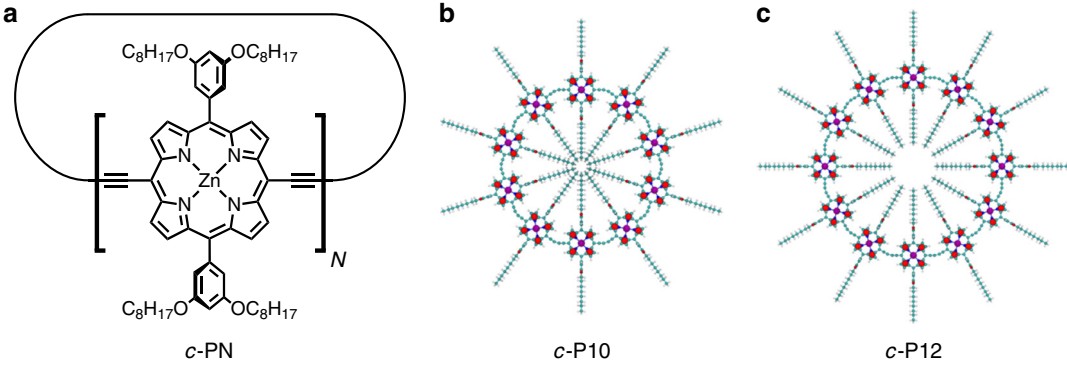

**Fig. 1** Structure of $c$-PN (cyclic nanoring with N porphyrin groups) porphyrin nanorings. **a** Generic structural formula of cyclic $c$-PN polymer, **b** atomistic models of $c$-P10 and **c** $c$-P12 used in molecular dynamics simulations

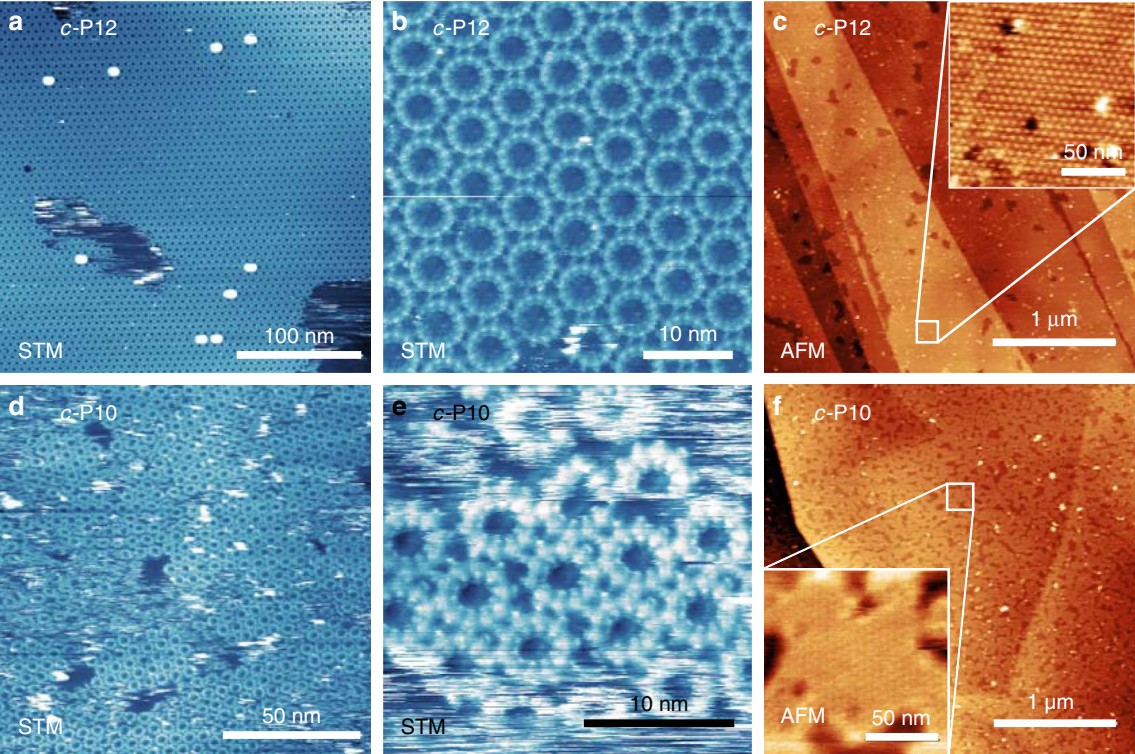

**Fig. 2** Scanning probe microscopy (SPM) images of porphyrin nanorings deposited on a highly oriented pyrolytic graphite (HOPG) surface. **a** Scanning tunnelling microscopy (STM) image of continuous *c*-P12 domain. The bright spots correspond to stacks of multiple *c*-P12 rings. **b** High-resolution STM image showing packing of *c*-P12 molecules in a single domain. **c** Atomic force microscopy (AFM) image showing large-scale structure of *c*-P12 domains on HOPG. Dark regions correspond to voids in the nanoring monolayer. (Inset) High-resolution image showing hexagonal packing of *c*-P12 molecules in region indicated by the white box in the main image. **d** STM image of *c*-P10 domains on HOPG. **e** High-resolution image of *c*-P10 molecules. **f** AFM image showing large-scale structure of *c*-P10 domains on HOPG. (Inset) High-resolution image of a single *c*-P10 domain at the region indicated by the white box in the main image

around the near-circular perimeter of each nanoring correspond to the component porphyrin macrocycles. Interestingly, we find that neighbouring nanorings are aligned so that pairs of porphyrin macrocycles meet along their common boundary (see schematic overlay in Fig. 3a, b). AFM images of dry films of *c*-P12 (see Fig. 2c) also show large single domains with hexagonal order, but the measured periodicity is slightly larger, $6.8 \pm 0.1$ nm, than observed in liquid conditions. As we discuss later, the larger measured periodicity for dry films is in excellent agreement with our simulation results, suggesting that the liquid conditions may cause a change in the conformation of the alkane side-groups with respect to the surface allowing a higher packing density[31].

The global alignment of porphyrin groups in neighbouring nanorings (Fig. 2b) is only compatible with the formation of a close-packed array with hexagonal order if $N$, the number of macrocycles, is a multiple of 6. This condition is violated for the *c*-P10 nanoring providing a route to explore the packing behaviour for structures where the symmetries of packing and the internal molecular structure are not compatible. Large area images (Fig. 2d–f) show that adsorbed arrays of *c*-P10 form domains with typical lateral dimensions ~50 nm, much smaller than those formed by *c*-P12 aggregates. In large-scale images (Fig. 2d), the *c*-P10 islands appear to be close to hexagonal, although somewhat disordered, and form two rotational configurations with an angle of ~10° between their principal axes. In addition, the *c*-P10 islands, unlike *c*-P12, are rather unstable leading to a streaky appearance of the STM images (Fig. 2d, e), consistent with mobile material on the surface, and prolonged STM imaging is not possible. The typical separation of the centres

of neighbouring nanorings is ~5 nm. AFM imaging (Fig. 2f) of dry *c*-P10 films on HOPG confirm the STM observations showing many small domains with a slightly higher characteristic nanoring separation of ~5.5 nm when compared to the liquid STM measurements in agreement with the behaviour of *c*-P12.

Higher magnification images (Fig. 2e) show that the *c*-P10 nanorings exhibit several different packing arrangements, which can be classified by the alignment of porphyrin groups within neighbouring nanorings. First, we see a distorted hexagonal arrangement (Fig. 3d) in which the pairs of porphyrins in neighbouring nanorings are aligned in a similar arrangement as observed for *c*-P12 along one principal axis. However, along the other two axes we find that single porphyrin groups are aligned in a row-like fashion (see dotted lines and schematic in Fig. 3c). We refer to this as a "row" phase and identify regions of the surface in this arrangement with the yellow dashed outline in Fig. 3g.

In *c*-P10 aggregates, we also see a distinct mode of packing in which four nanorings combine to form a rhombus shape as shown in Fig. 3e, f with approximate internal angles of 70° and 110°. This is not a close-packed phase, but is formed from junctions of nanorings in which pairs of porphyrins on neighbouring nanorings are aligned; these junctions are similar to those observed for *c*-P12. The single unit cell of the rhombic phase in Fig. 3f connects two domains of the row phase, and the pair-wise alignment of porphyrin groups is indicated by the white dashed line. A region with this arrangement repeated over a small area is highlighted by the black dashed outline in Fig. 3g. This rhombic order has been predicted in theoretical studies of assembly of rigid discs with 10-fold symmetric interparticle

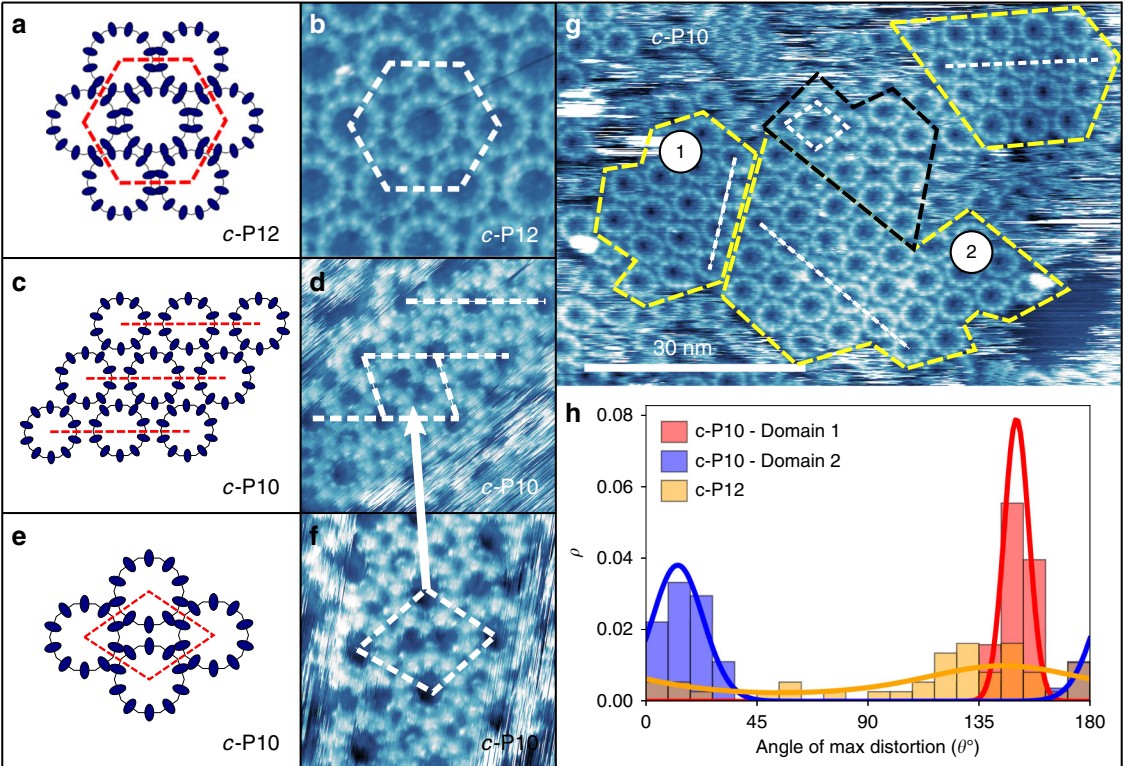

**Fig. 3** Scanning tunnelling microscopy (STM) images showing the arrangement of *c*-P10 and *c*-P12 nanorings on highly oriented pyrolytic graphite (HOPG). **a** Schematic showing the hexagonal packing arrangement observed for *c*-P12. **b** STM image of *c*-P12 hexagonal packing. **c** Schematic showing the relative orientation of porphyrin groups in the row packing phase for *c*-P10. **d** STM image of *c*-P10 row-like packing connected by a single unit cell of the rhombic phase shown in **f** and indicated by the white arrow. **e** Schematic of *c*-P10 rhombic phase. **f** STM image of *c*-P10 rhombic packing connecting two domains of the row phase shown in **d**. The white dashed lines in images **b**, **d**, **f** indicate the direction of double porphyrin-pair alignment shown by the red dashed lines in the schematic images **a**, **c**, **e**. **g** STM image of multiple *c*-P10 domains on HOPG showing regions with row (yellow outline) and rhombic (black outline) packing arrangements. The dashed white lines indicate the directions of double porphyrin–porphyrin alignment shown for both the row and rhombic arrangements. **h** Histograms showing the direction of maximum distortion of the *c*-P12 domain shown in Fig. 2b (orange) and the two labelled *c*-P10 domains in Fig. 3g (red and blue). The solid lines are fits to the data using a von Mises distribution of the circular mean and standard deviation

interactions[16]; flexibility and isotropic (van der Waals) interactions were not considered in this model and likely prevent the emergence of long-range rhombic order.

Interestingly, we also observe that the *c*-P10 nanorings in the row phase are slightly distorted into an elliptical shape with the short axis of the distorted nanoring parallel to the direction of the row alignment of *c*-P10. To be consistent with our earlier investigations of distortion exhibited by isolated nanorings[6,17], the degree of distortion from a circular conformation is characterised using the parameter $\bar{g} = a/b - 1$, where $a$ and $b$ are the long and short axes of the nanoring, respectively. The values for $a$, $b$ and also $\theta$, the angle of the major elliptical axis relative to a fixed (horizontal) axis, were extracted by computationally fitting ellipses to the positions of the porphyrin monomers around the ring.

The distortion of nanorings is summarised in a histogram (Fig. 3h) of the nanoring rotation angles and a table (Table 1) of the average values of $\bar{g}$ for the two numbered *c*-P10 domains in Fig. 3g (red/blue histograms) and the *c*-P12 rings shown in Fig. 2b (orange histogram). The histograms clearly exhibit a much narrower angular distribution for the *c*-P10 rings, confirming the visual impression that the nanorings are not only distorted, but that the deformation is correlated across each *c*-P10 domain. In particular, the orientation of the major elliptical axes of the nanorings with respect to the geometry of their packing (the row axis) is fixed within each domain. This result suggests that the nanorings undergo distortion in a co-operative manner with

energy of deformation compensated by nearest-neighbour interactions which stabilise the locally ordered array.

The amplitude of the distortion, quantified by $\bar{g}$, is approximately 0.2 for the *c*-P10 domains (see Fig. 3 and Table 1). In comparison, however, the average distortion of the *c*-P12 rings is much less, $\bar{g} = 0.06 \pm 0.01$, and, in addition, the distribution of the orientation of the major axes is much broader. We argue that this behaviour is consistent with an array of near-circular nanorings, which exhibit small, uncorrelated shape fluctuations. These results are consistent with the visual appearance of *c*-P12 (see Fig. 3) where there is no obvious preferred directionality to the nanoring distortion.

**Numerical studies of nanoring adsorption.** The local ordering of *c*-P10 and *c*-P12, the stabilisation effects due to nanoring

---

**Table 1 Average distortion value, $\langle\bar{g}\rangle$, for nanorings in different domains**

| Domain | $\langle\bar{g}\rangle$ |
|---|---|
| *c*-P10-1 | $0.22 \pm 0.02$ |
| *c*-P10-2 | $0.18 \pm 0.01$ |
| *c*-P12 | $0.06 \pm 0.01$ |

Data are shown for the *c*-P10 row domains labelled 1 and 2 in Fig. 3g and for the *c*-P12 domain in Fig. 2b

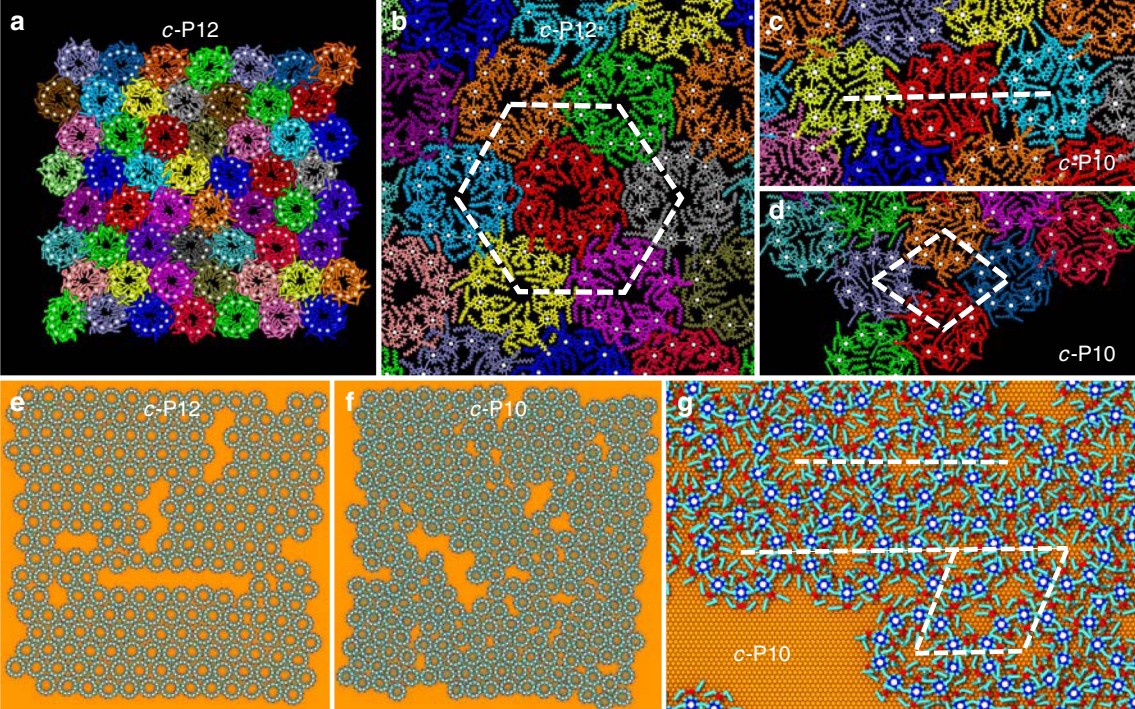

**Fig. 4** Simulation snapshots of c-P10/12 nanorings. **a** Snapshot from an atomistic molecular dynamics (MDs) run of c-P12 aggregate. **b** Hexagonal arrangement of c-P12 nanorings. **c** Row and **d** rhombic packing of simulated c-P10 nanorings. **e–g** Snapshots taken from CG MD run of **e** c-P12 and **f** c-P10 aggregates. **g** Example of contiguous row and rhombic motif in c-P10 aggregate

deformation, and the intermolecular interactions in the neighbouring rings have been further explored using classical MD and coarse-grained (CG) simulations. MD simulations have been performed using the LAMMPS simulation package[32], in which the well-established OPLS (optimized potentials for liquid simulations) potential[33] has been employed with additional parameters derived from density functional theory calculations as implemented in the CP2K program package[34]. Further details of the computational setup are given in Supplementary Methods.

In our MD simulations, the c-P12 aggregates (represented by 56 nanorings with the total number of atoms reaching 250,000) show almost perfect, thermally stable, hexagonal packing (see Fig. 4a, b where porphyrin groups are indicated by the white dots) exhibiting the alignment of neighbouring porphyrin pairs in excellent agreement with the STM results (Fig. 3b). Away from the edges of domains, the average attractive energy of interaction between neighbouring nanoring pairs is calculated to be 1.2 eV (see Supplementary Fig. 9b). Typically 20–30% of this energy is due to the interaction of alkane sidechains as shown in Supplementary Fig. 9c. In addition, there is an interaction energy between the sidechains and the substrate, which is calculated to be, on average, 0.34 eV/sidechain. The role of the sidechains in stabilising nanorings is consistent with experimental observations that nanorings with much smaller[18] (tertiary butyl) sidechains do not form stable arrays on the surface (see Supplementary Fig. 2). The two-dimensional radial distribution function corresponding to the centre of mass of Zn atoms in each nanoring has a sharp dominant peak at 6.8 nm (see Supplementary Fig. S9a) in excellent agreement with the AFM observations for the c-P12 aggregates shown in Fig. 2c.

In the MD simulations, as in experiments, the c-P10 aggregates are less ordered than c-P12 and the neighbouring rings show a variety of rotational arrangements. Far from the aggregate edges we observe spontaneous formation of small row packing domains as shown in Fig. 4c. Rhombic motifs also appear in various

regions of the aggregate; these are readily formed at the edges (see Fig. 4d) where nanorings are less constrained by the neighbours and can rearrange their position during the MD simulation time of 200 ns (with the time step of 1 fs). In agreement with the STM observations, c-P10 aggregates are found to be mobile and exhibit continuous structural reconstructions (see Supplementary Fig. 8 for further details), while c-P12 aggregates remain essentially stable once fixed into the hexagonal arrangement.

In addition to the detailed large-scale atomistic simulations, we have used a CG approach to model a larger number of rings, up to 224 in total, and to increase simulation time to 400 ns (using a 15 fs time step). The MARTINI CG potential[35,36] has been employed with additional parameters fitted to the MD simulations (see SI for further details). Figure 4 also shows simulation snapshots of the GC MD model for c-P12 (Fig. 4e) and c-P10 (Fig. 4f) aggregates. The difference in both short- and long-range order is clearly visible. In both cases, because of the initial large spacing between the nanorings, some voids in the nanoring aggregate remain present. Apart from a few defects between the well-formed islands, the c-P12 aggregate shows an ordered hexagonal packing, which is consistent across the void regions. The c-P10 aggregate is much more disordered with neighbouring rings exhibiting a variety of motifs and, at longer range, islands of different orientational alignments. As shown in Fig. 4g, it is possible to observe the formation of row and rhombic motifs. The two-dimensional radial distribution function corresponding to the centre of mass of Zn atoms in c-P10 can be found in the SI for both MD structures (Supplementary Fig. 9a; peak value 5.7 nm) and CG structures (Supplementary Fig. 11), providing a good match with the experimental values. The c-P12 aggregates show sharp peaks almost resembling a crystalline structure, whereas for c-P10 they coalesce into a broader shoulder indicating a less ordered aggregate at longer range, which is consistent with various domains of different morphology. The small island sizes, particularly for c-P10, preclude an analysis of correlations in

deformation, although the average values for distortion, ḡ, can be determined for the simulated nanorings. The distortion parameter for simulated c-P12 is ḡ = 0.08, close to the experimental value, while for c-P10, ḡ = 0.1; this is smaller than the experimental value, possibly due to the small domain size of the simulated structures, although the trend of higher distortion for c-P10 observed experimentally is reproduced in simulations. Overall there is excellent agreement between experiment and theory in the spacing of nanorings, alignment of neighbouring groups and the appearance of the basic packing arrangements.

Different motifs arise from a balance of non-directional interactions such as van der Waals interactions, which always favour a maximal packing density, the elastic energy of deformation of the nanorings, and the quasi-directional interactions resulting from the interactions between porphyrins on neighbouring nanorings, which appear to favour junctions where these groups are aligned. For c-P12 these driving forces can be simultaneously minimised, but due to geometric constraints this is not possible for c-P10 and we observe a frustrated packing resulting from a competition between these effects. Specifically, in the row phase the packing is greater, but the junctions between neighbouring nanorings is non-optimal resulting in an elastic deformation of the nanorings, while in the rhombic phase the inter-nanoring interactions are minimised, but there is a lower packing density. The co-existence of these packing arrangements on the surface implies that they must have a similar energy; this would also account for the higher disorder and smaller domain sizes observed (experimentally and in calculations) for c-P10.

Our calculations also allow a comparison of the relative energies involved in close packing and distortion of nanorings. As shown in Table 1, the measured values of ḡ for the c-P10 nanorings exhibiting the row packing phase are ḡ = 0.22 ± 0.02 and 0.18 ± 0.01 for the domains labelled 1 and 2, respectively, in Fig. 3g. The energy required to distort c-P10 nanorings with these levels deformations are 0.38 eV (domain 1) and 0.24 eV (domain 2), much smaller than the inter-nanoring stabilisation energies (1.2 eV). This supports our hypothesis that the nanoring distortion in the frustrated phases is compensated by increased inter-nanoring interactions. Interestingly, a similar calculation for c-P12 with the experimentally observed average value of ḡ = 0.06 yields a distortion energy of 4 meV. This value is close to the limits of accuracy of our calculations for the deformation energies, but we note that it is lower than the thermal energy at room temperature (30 meV) providing additional confidence in our conclusion above that the observed deviations from circularity within the c-P12 arrays are due to random fluctuations rather than co-operative behaviour.

## Discussion

Our observations of c-P10/12 nanorings show clearly that changes in the internal symmetry of the deformable c-PN species has a significant effect on the packing behaviour of these molecules, for example, in the dramatic increase in the domain size and stability of c-P12, when compared to c-P10, which adopts a frustrated packing arrangement. This behaviour is replicated using both full MD and CG simulation methods. These results reveal the emergence of certain geometric motifs, such as rhombic motifs in c-P10 arrays, that can be understood from simple principles of symmetry[16], while the addition to this picture of molecular detail and flexibility accounts for the diversity of structures we observe. Our results demonstrate that fundamental studies of deformable molecular systems on surfaces can provide insights into complex behaviour in which the shapes of individual adsorbed species can responsively adapt to the presence of nearest neighbours giving rise to new, emergent modes of structural organisation of

macromolecular systems, including those of relevance to technological applications and biological materials[25,26].

## Methods

**STM/AFM imaging of c-P10/12 nanorings.** c-P12 and c-P10 powders were synthesised[19,20] and then dissolved in a 3:1 toluene:methanol (tol:MeOH) (Sigma-Aldrich) mixture to produce a 100 µg/ml stock solution and stored at 5 °C until required. For deposition onto HOPG, the stock solutions were further diluted 1:9 in toluene to give a 10 µg/ml in 39:1 tol:MeOH solution for STM and AFM experiments. HOPG substrates (Agar Scientific) for STM and AFM imaging were prepared by exfoliating the top layers with adhesive tape to cleave the surface. Nanorings were deposited onto the substrate for STM imaging by placing a 20 µl drop of the diluted nanoring solution onto the clean HOPG surface for 10–30 min under a cover to prevent rapid evaporation. After the required time, substrates were removed from the solution and dried rapidly in a stream of N₂.

Following deposition of nanorings from solution, the substrate was then mounted onto the STM stage and a 10 µl droplet of nonanoic acid (Sigma-Aldrich) was deposited onto the sample for imaging. Pt:Ir (80:20%) wire was cut and used as an STM tip. STM images were acquired using a Molecular Imaging (Agilent) PicoSTM unit in constant current mode. Typical scanning parameters were: +0.5–1 V applied to the tip and a constant tunnelling current of 20–50 pA for imaging of c-P10/c-P12 monolayers. For imaging the underlying HOPG substrate in Supplementary Fig. 1b, c; +0.1 V and 1 nA, respectively, were used as scanning parameters.

For ambient AFM imaging, the substrates were prepared identically to the STM substrates, omitting the addition of nonanoic acid before imaging. The substrates were then mounted in an Asylum Research Cypher-S AFM system and imaged in repulsive tapping AC mode using Olympus AC240TS silicon cantilevers driven just off resonance ($F_0$ = 70 kHz, $k$ = 2 N/m) at set points between 60 and 70% of the free-air amplitude. All STM and AFM images were processed using the Gwyddion software package[37]. It was found that the coverage of nanorings was not strongly dependent on the deposition time for times >10 min. It was also found from further STM/AFM experiments that the omission of MeOH in the stock nanoring solution did not have an effect on the deposition of c-P10/c-P12 monolayers on HOPG.

## Data availability

The raw data for the AFM and STM images may be accessed through the University of Nottingham Research Data Management Repository at https://doi.org/10.17639/nott.6997.

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

## Acknowledgements

This work was supported by the Engineering and Physical Sciences Research Council [grant numbers EP/J006939/1, EP/P019080/1 and EP/J007161/1]; the Leverhulme Trust [grant number RPG-2016-104]; and the European Research Council (grant 320969). D.V. K. thanks the Clarendon Fund of the University of Oxford for a studentship. S.W. performed work at the Molecular Foundry, Lawrence Berkeley National Laboratory, supported by the Office of Science, Office of Basic Energy Sciences, of the US Department of Energy under Contract No. DE-AC02–05CH11231.

## Author contributions

A.S., P.H.B and H.L.A. conceived the experimental project with further contributions from D.V.K.; the STM and AFM imaging were carried out by A.S.; H.L.A. and D.V.K. synthesised the nanorings; the analysis of domain structure was performed by A.S., P.H.B., J.P.G., S.W., M.B. and E.B.; the DFT, MDs and coarse-grained calculations were carried out by M.B. and E.B.; A.S., P.H.B., M.B. and E.B. wrote the paper with revisions and comments from all authors.

## Additional information

**Competing interests:** The authors declare no competing interests.

