## [Peer Review File · Nature Communications]

Reviewers' comments:

Reviewer #1 (Remarks to the Author):

This paper describes the two-dimensional packing of cyclic porphyrin polymers, porphyrin nanoring. Although the content is an interesting topic in the chemical and related fields. the similar research contents regarding the observation of molecular and supramolecular structures were widely reported (e.g., reviewer paper: Chem. Commun., 2016,52, 11465-11487). Therefore I'd like to resubmit this manuscript to a more specialized journal in the field of chemistry.

Reviewer #2 (Remarks to the Author):

This manuscript describes an experimental and computational study of the self-assembly of porphyrin nanorings with different symmetry into close-packed structures on a graphite surface. With STM and AFM, the authors demonstrate that while c-P12 nanorings assemble into highly regular hexagonal close-packed lattices, c-P10 nanorings display more disordered patterns. The authors show that the disruption of regular hexagonal symmetry is due to a mismatch between the internal 10-fold symmetry of nanorings and the 6-fold symmetry of the hexagonal lattice. They go on to identify two distinct packing patterns of c-P10, which coexist in experimental c-P10 nanoring patterns. Atomistic and coarse-grained simulations are performed that essentially confirm the experimental results.

Understanding the competing driving forces that lead to ordered and disordered self-assembled patterns is an important goal of nanoscience and a prerequisite for controlled assembly of target structures for applications. This manuscript describes a particularly neat example of molecular self-assembly which displays frustrated attractions that cause interesting pattern formation. The experimental work allows a detailed and unambiguous characterization of monomers and self-assembled structures. Together with the simulations, substantial new insight is gained into the competition of molecular structure and packing. The manuscript is well written and will be of broad interest to the scientific community. I recommend publication in Nature Communications after the following comments have been addressed.

1. While the experimental images identify the "double-porphyrin" binding mode between nanorings unambiguously, the authors do not analyze the molecular interactions that are responsible for this binding mode, which should be possible from their atomistic simulations. Is nanoring-nanoring binding primarily due to interactions between alkane chains? If yes, could patterns be tuned by changing the length of the alkane chains?
2. The authors point out similarities with a computational study by Whitelam and coworkers (Ref. 16), which finds, for discs with 10 patches that appear to be a reasonable model for c-P10 nanorings, a similar competition between hexagonal and rhombic patterns. Interestingly, Ref. 16 predicts a phase transition from primarily rhombic to primarily hexagonal as a function of patch width. The authors should give more insight into the similarities/differences of their system and the simple model in Ref. 16. Can they demonstrate a similar switching between patterns (perhaps as a function of alkane chain length)?
3. In Fig. 4, the authors should show a full view of the c-P10 simulation, analogous to panel (a).
4. I am worried about the ability of the simulations to access equilibrated configurations. The authors point out that interactions between nanorings are strong (1.2 eV), which suggests the presence of potentially large barriers for rotation of nanorings. Can the authors confirm that the "double-porphyrin" binding between nanorings observed in simulations is not simply a consequence of their regular initial configuration? Have the authors tried to anneal the structures at higher temperatures?
5. A perhaps related question: Why do the authors not observe larger domains of rhombic order (as observed in experiments, Fig. 3g) in simulations?

6. The author present an analysis of different levels of distortion of nanorings in different types of patterns. The authors should report if similar distortions are observed in the simulations.

Reviewer #3 (Remarks to the Author):

The authors present a comprehensive experimental and theoretical study of the self-assembly of flexible porphyrin nanorings.

It is clearly observed and justified by simulations that changes in the internal symmetry of the deformable porphyrin nanorings have a significant effect on the packing behavior of these molecules.

The nanoring of size 12 is a multiple of 6 and thus is compatible with the formation of a close-packed array with hexagonal order, as observed. Herein, deformation of the species is low and assigned to random fluctuations.

The nanoring of size 10 provides a route to explore the packing behavior for structures where the symmetries of packing and the internal molecular structure are not compatible. Consequently, here they find frustration since close packing and co-alignment of neighbouring of neighbouring groups cannot be simultaneously realized. Concomitantly, they detected distortion of the nanorings into elliptical shapes. The authors hypothesize that the nanoring distortion in the frustrated phases is compensated by increased inter-nanoring interactions.

The manuscript is very well written, the experiments are nicely performed and the results are sound and important for the community. Thus, I would strongly recommend publication.

I have just some very minor comments:

- * Have the authors considered to measure LEED of the samples?
- * Have the authors tried to analyze the FFT of their images?

Both suggestions could reinforce their analysis of the growth domains.

Response to Referees: NCOMMS-19-05728-T

We thank the referees for their comments and we have modified our paper in response to their suggestions as described below

Reviewer #1 (Remarks to the Author):

This paper describes the two-dimensional packing of cyclic porphyrin polymers, porphyrin nanoring. Although the content is an interesting topic in the chemical and related fields. the similar research contents regarding the observation of molecular and supramolecular structures were widely reported (e.g., reviewer paper: Chem. Commun., 2016,52, 11465-11487). Therefore I'd like to resubmit this manuscript to a more specialized journal in the field of chemistry.

The paper mentioned by the referee is a review covering some of the recent significant advances in surface supramolecular structures. This review highlights the very diverse fields of investigations which can be explored in these layers but so far these studies have not encompassed the topic of our paper which is the relationship between flexibility, packing and order. We argue that our work therefore complements these existing studies by demonstrating a new phenomenon. We disagree with the referee that the interest in our results will be limited to the chemistry community; in fact the problem of packing objects on a surface is much broader with implications for crystallography and statistical physics. Furthermore, the nanorings which we study have similarities to biologically important molecules involved in light harvesting complexes in photosynthetic membranes. We therefore argue that our paper is very well suited to the interdisciplinary emphasis of Nature Communications. We have made several changes to the paper to strengthen the interdisciplinary aspects and have added several references including the paper mentioned by the referee and, in addition, some related to photosynthetic complexes in the introduction and conclusion sections.

Reviewer #2 (Remarks to the Author):

This manuscript describes an experimental and computational study of the self-assembly of porphyrin nanorings with different symmetry into close-packed structures on a graphite surface. With STM and AFM, the authors demonstrate that while c-P12 nanorings assemble into highly regular hexagonal close-packed lattices, c-P10 nanorings display more disordered patterns. The authors show that the disruption of regular hexagonal symmetry is due to a mismatch between the internal 10-fold symmetry of nanorings and the 6-fold symmetry of the hexagonal lattice. They go on to identify two distinct packing patterns of c-P10, which coexist in experimental c-P10 nanoring patterns. Atomistic and coarse-grained simulations are performed that essentially confirm the experimental results.

Understanding the competing driving forces that lead to ordered and disordered self-assembled patterns is an important goal of nanoscience and a prerequisite for controlled assembly of target structures for applications. This manuscript describes a particularly neat example of molecular self-assembly which displays frustrated attractions that cause interesting pattern formation. The experimental work allows a detailed and unambiguous characterization of monomers and self-assembled structures. Together with the simulations, substantial new insight is gained into the competition of molecular structure and packing. The manuscript is well written and will be of broad interest to the scientific community. I recommend publication in Nature Communications after the following comments have been addressed.

1. While the experimental images identify the "double-porphyrin" binding mode between nanorings unambiguously, the authors do not analyze the molecular interactions that are responsible for this binding mode, which should be possible from their atomistic simulations. Is nanoring-nanoring binding primarily due to interactions between alkane chains? If yes, could patterns be tuned by changing the length of the alkane chains?

We have extracted further details from the atomistic simulations to show that a substantial, but not dominant, contribution to the nanoring - nanoring binding energy (typically 20-30%) comes from the direct interactions between alkane chains; our analysis also shows that the chains contribute to the interaction with the substrate (0.34 eV/chain corresponding to 16.3 eV/nanoring for *c*-P12). Figure S9 has been modified and expanded and some additional text added to the SI to clarify this point and a comment and cross-reference has been added to the main paper. The effect of the length of alkane chains is addressed in the next query.

*2. The authors point out similarities with a computational study by Whitelam and coworkers (Ref. 16), which finds, for discs with 10 patches that appear to be a reasonable model for *c*-P10 nanorings, a similar competition between hexagonal and rhombic patterns. Interestingly, Ref. 16 predicts a phase transition from primarily rhombic to primarily hexagonal as a function of patch width. The authors should give more insight into the similarities/differences of their system and the simple model in Ref. 16. Can they demonstrate a similar switching between patterns (perhaps as a function of alkane chain length)?*

Experimentally the investigation of this interesting question would require through the synthesis of nanorings with different sidegroups. However, there are severe practical limitations to the synthesis of many analogue cyclic polymers. This is partly due to the role of the sidegroup which is included to promote solubility in the solvents which are used in the synthetic process and, while, in principle the chain length can be changed this can generate significant complexities since the processes are optimised for specific solvent conditions. This is compounded in the present case by the fact that the synthetic process is already highly complex through the use of a molecular template and Vernier scaling technique to generate the cyclic polymers used in this study; this synthetic route is thus extremely demanding and time consuming and it is possible to investigate only a small number of variants, thus precluding a systematic study of chain length. Nevertheless, we have investigated the adsorption of *c*-P12 nanorings with a much smaller solubilising group (tertiary butyl groups are used as an alternative to the alkoxy chains) for which a synthetic pathway has been established. For this variant we observe only small numbers of molecules and these are adsorbed at rather obvious defects, such as step edges, and we do not observe any extended close-packed islands in this case. Consequently, it has not proved possible to undertake systematic measurements with this unstable analogue. We have included an additional figure in SI (Figure S12) showing the images of the variant with tertiary butyl solubilising groups and added an appropriate cross-reference in the main text together with a brief statement related to the previous point, i.e. the role of the alkane chains.

*3. In Fig. 4, the authors should show a full view of the *c*-P10 simulation, analogous to panel (a).*

We have added the suggested figure, but prefer to place in the SI (Fig. S8b). Some of the well-formed rhombic and row-like packing features are highlighted with dashed white lines.

4. I am worried about the ability of the simulations to access equilibrated configurations. The authors point out that interactions between nanorings are strong (1.2 eV), which suggests the presence of potentially large barriers for rotation of nanorings. Can the authors confirm that the “double-porphyrin” binding between nanorings observed in simulations is not simply a consequence of their regular initial configuration? Have the authors tried to anneal the structures at higher temperatures?

We have further tested the stability of the obtained equilibrium configurations by annealing both the *c*-P10 and *c*-P12 structures at 400K for 50 ns. Snapshots taken from MD runs show that *c*-P12 aggregates remain substantially unchanged whilst the *c*-P10 structure exhibits a continuous reconstruction, in agreement with observations at room temperature. Figure S8 has been

substantially modified and expanded to include snapshots from atomistic MD runs of c-P10 and c-P12 nanorings annealed at 400K. Some additional text was also added to the SI.

5. A perhaps related question: Why do the authors not observe larger domains of rhombic order (as observed in experiments, Fig. 3g) in simulations?

We acknowledge the fact that the rhombic order is observed over larger areas in the images, but even these domains are rather small and are limited to 2-4 lattice constants. In fact, we were rather encouraged to find that similar motifs occur at all in both theory and experiment. It is possible that this difference may be due to finite size effects since in both experiment and theory many of the c-P10 nanorings are in rather small domains – significantly different (in both theory and experiment) to the c-P12.

6. The author present an analysis of different levels of distortion of nanorings in different types of patterns. The authors should report if similar distortions are observed in the simulations.

We observe a similar trend between the average distortion of c-P12 and c-P10 in the simulations and we have added a statement related to this in the main text. It has not proved possible to analyse the simulated arrays to determine the effects of correlation for the reasons discussed in the previous point, i.e. that the domain sizes are small, particularly for c-P10, in the simulations meaning that edge effects are more significant.

Reviewer #3 (Remarks to the Author):

The authors present a comprehensive experimental and theoretical study of the self-assembly of flexible porphyrin nanorings.

It is clearly observed and justified by simulations that changes in the internal symmetry of the deformable porphyrin nanorings have a significant effect on the packing behavior of these molecules.

The nanoring of size 12 is a multiple of 6 and thus is compatible with the formation of a close-packed array with hexagonal order, as observed. Herein, deformation of the species is low and assigned to random fluctuations.

The nanoring of size 10 provides a route to explore the packing behavior for structures where the symmetries of packing and the internal molecular structure are not compatible. Consequently, here they find frustration since close packing and co-alignment of neighbouring of neighbouring groups cannot be simultaneously realized. Concomitantly, they detected distortion of the nanorings into elliptical shapes. The authors hypothesize that the nanoring distortion in the frustrated phases is compensated by increased inter-nanoring interactions.

The manuscript is very well written, the experiments are nicely performed and the results are sound and important for the community. Thus, I would strongly recommend publication.

I have just some very minor comments:

** Have the authors considered to measure LEED of the samples?*

We have not measured LEED. We are aware from our previous work that the nanorings are rather delicate – they are broken when annealing to ~ 100 °C and we would anticipate that beam damage would be a problem; furthermore this vacuum-based technique is not straightforwardly compatible with the solution deposition which we use. The resolution using AFM and STM of nanorings is not a significant problem and we took extensive steps to eliminate the effects of drift etc. as described in the SI.

** Have the authors tried to analyze the FFT of their images?*

We have used FFT analysis at some points but were not able to identify any additional periodic structure which was not already evident in the real-space images.

REVIEWERS' COMMENTS:

Reviewer #2 (Remarks to the Author):

The authors have addressed the points raised by this reviewer. I recommend publication.

Reviewer #3 (Remarks to the Author):

The authors have satisfactorily replied point to point to all the questions from the referees.

Thus I would strongly recommend publication of the manuscript as it is.